# Alteration of Flower Yield and Phytochemical Compounds of Saffron (*Crocus sativus* L.) by Application of Different Light Qualities and Growth Regulators

**Mostafa Eftekhari** [1], **Majid Ghorbani Javid** [1,*], **Sasan Aliniaeifard** [2] **and Silvana Nicola** [3,*]

1    Department of Agronomy and Plant Breeding Sciences, College of Aburaihan, University of Tehran, Tehran P.O. Box 3391653755, Iran
2    Photosynthesis Laboratory, Department of Horticulture, College of Aburaihan, University of Tehran, Tehran P.O. Box 3391653755, Iran
3    Department of Agricultural, Forest, and Food Sciences—DISAFA, Vegetable Crops and Medicinal & Aromatic Plants, University of Turin, 10095 Grugliasco, Italy
*    Correspondence: mjavid@ut.ac.ir (M.G.J.); silvana.nicola@unito.it (S.N.)

**Abstract:** Saffron is the world's most coveted spicy plant that has medicinal value. Currently, due to diverse types of difficulties in growing this plant outdoor, the tendency to produce it indoor has been increased. Optimized indoor conditions for growing saffron plants is not fully determined so far. This study was conducted to investigate the interactive effects of two plant growth regulators (PGRs), including gibberellic acid ($GA_3$) and γ-aminobutyric acid (GABA) and four light recipes, including white, monochromatic blue, monochromatic red, and a combination of 50% red and 50% blue on the flower yield and phytochemical components (such as crocin, picrocrocin and safranal) in stigmas of indoor-grown saffron. The results showed that exogenous GABA application and combined red and blue LED lights enhanced the performance of saffron flowers in terms of the number of flowers (up to 1.97 per corm) as well as the fresh and dry weight of flowers and stigmas. In saffron, the concentration of three major secondary metabolites is of great importance since it determines its commercial, pharmaceutical quality. GABA induced saffron's chemical ingredients toward the phytochemicals safranal (up to 5.03%) and picrocrocin (up to 15.8%), while $GA_3$ induced them toward the carotenoid pigment crocin (up to 25.1%). In conclusion, the application of GABA with a combination of red and blue lights enhanced the production of high-quality stigmas and positively affected the yield of flowers in saffron plants.

**Keywords:** flowering; growth regulators; indoor farming; light quality; phytochemical composition

## 1. Introduction

Saffron is the world's most coveted, lucrative spice, which is mainly derived from the stigmas of the saffron (*Crocus sativus* L.) flower. It is used for flavoring, fragrance, dyes and medicines. Saffron's taste and iodoform-like or hay-like fragrance result from the phytochemicals such as picrocrocin and safranal [1,2]. Likewise, it contains a carotenoid pigment, crocin, which imparts a rich golden-yellow hue to dishes and textiles. The global yield of saffron under open field conditions is estimated at 3.4 kg ha$^{-1}$ (418 t y$^{-1}$ of production in 121,338 ha) [3]. While some doubts remain on its origin [4], it is believed that saffron originated from Iran [5]. Iran currently produces 90% of the world's saffron [6,7].

Saffron is still produced traditionally in arid and semi-arid regions [8]. Producing saffron outdoors faces many difficulties such as limited yield due to the negative impacts of climate change, most importantly increased temperatures. Modern technologies using indoor culture methods have not been widely used for the production of saffron, despite the antiquity of its cultivation [9]. Controlled-environment agriculture (CEA), which includes indoor agriculture (IA) and vertical farming, is a technology-based approach that

recently attracted the attention of saffron production. The CEA aims to provide protection from the outdoor constraints and to maintain proper growing conditions throughout the development of the crop. Production takes place within an enclosed growing structure such as a greenhouse, vertical farm, or indoor grow room [10]. The positive effects of light emitting diode (LED) lamps on saffron cultivation have been shown in a LED-equipped controlled-environment room [11]. CEA optimizes the use of resources such as water, energy, land and space, capital investment and labor [12], and it paves the way toward creating the optimal growing conditions for saffron production. CEA aims to optimize plant growth mainly through using soilless farming techniques such as hydroponics and aeroponics techniques [13]. Fogponics is an advanced form of aeroponics that uses some specific mechanisms (for example ultrasonic, compressed air, or heating elements) to form a suspension of much smaller particles of nutrient enriched water (5–30 μm), or even as a vapor [14,15]. The ultrasonic fog generator capable of vibrating at supersonic frequencies and produced micro droplets of water nutrient make it easier and faster for the plant to absorb via the roots.

Aghhavani-Shajari and collaborators compared the flowering of saffron between an open field and a controlled environment [16]. They reported that, in the first and the second flowering seasons, the percentage of flowering corms was higher under the soilless production system (39 and 65%, respectively) than the traditional production in soil (6 and 56%, respectively). Even though saffron flowers can be simply produced through indoor farming methods, the production of high-quality saffron through indoor farming methods is still challenging. Flower initiation and emergence in saffron are sensitive to environmental conditions, such as humidity, temperature and light [17]. Light is a key factor affecting phytochemical synthesis and accumulation in plants [18]. Over the past few years, light quality (i.e., the spectral specificity of light) has been remarkably considered as an important environmental factor in improving plant growth and quality in the indoor production of plants. Recent development in artificial lighting technology increases its use in indoor cultivation systems [19]. Current applications of vertical farming coupled with other advanced technologies, such as specialized LED lights, have resulted in over 10 times higher crop yields than traditional farming methods [20]. Numerous studies have investigated the growth and physiology of various plant species under red and blue lights [21–24]. LEDs with specified wavebands of light can be an easy tool to understand how the light spectrum influences plant growth and physiology. Red and blue spectra are the most effective lights in plant growth and development, especially for saffron production [11].

A large number of related chemical compounds are synthesized by humans and are used to regulate the growth and development of cultivated plants. These manmade compounds are called plant growth regulators (PGRs) that are used in many different techniques. PGRs play important roles in plant growth and development [25] owing to their important regulating role in plant physiology and biochemistry [26]. PGRs are one of the factors influencing essential oil production [27]. A four-carbon non-protein amino acid, $\gamma$-aminobutyric acid (GABA), is a significant component of the free amino acid pool in most prokaryotic and eukaryotic organisms [28]. GABA is also found in plants [29,30]. It has been shown that GABA controls a broad range of biological and physiological processes (such as C and N metabolisms, plant-microbe interactions, signal transduction and stress-derived responses) during plant growth and development [31]. Evidence also suggests a role in cell signaling in plants [32,33]. Gibberellic acid ($GA_3$) is a hormone found in plants and fungi [34]. It is also produced industrially using microorganisms [35]. Since $GA_3$ regulates growth, applications of very low concentrations (less than 250 ppm, according to Keshtkar and collaborators [36]) can have a profound effect while too much will have the opposite effect [37]. Gibberellins have a number of effects on plant development. They can stimulate rapid stem and root growth, induce mitotic division in the leaves of some plants, and increase seed germination rates [38]. It has been shown that the application of

GA$_3$ stimulated bud growth, increased flowering, and promoted the number and weight of flowers and yield of saffron [39].

The effect of light quality on the optimization of phytochemical concentration in plant species has been indicated in many studies [40–45]. The lighting spectra has been shown to affect the flower production of saffron through altering biomass partitioning toward flowers [11]; however, there is limited knowledge about the optimal lighting environment for saffron growth [46], particularly on the effect of light quality on saffron quality. On the other hand, the effects of indoor farming have not been widely studied on the flower yield and quality in bulbous plants such as saffron. Little is known about the role of exogenously applied PGRs in the yield and stigma quality of the saffron plant. Meanwhile, the interaction of PGRs and different light spectra on the amount of essence and the level of its composition in saffron has not been addressed so far. Given the information provided above, the present experiment was conducted to evaluate flowering and phytochemical composition in fogponically-grown saffron exposed to different PGRs and various light qualities. This research mainly aimed to address whether or how an optimal light quality and an ideal PGR can increase the yield and quality of saffron in the controlled environment. Specifically, we envisaged that, during the flowering period, an optimal light spectrum with an ideal PGR would have a synergistic effect, and subsequently, saffron yield and quality would increase.

## 2. Materials and Methods

### 2.1. Plant Materials and Growth Conditions

To evaluate the impacts of two PGRs and four different light qualities on saffron corms, this research was carried out as a two-factor factorial experiment in a randomized complete block design with three replications. The experimental factors included PGR at three levels (GA$_3$, GABA, control (just tap water) and light quality at four levels (W = white, B = blue, R = red, BR = combined blue and red) (Table S1). Saffron corms (*Crocus sativus* L.) were chosen from an Iranian ecotype called Qayen, which is native to the South Khorasan province (the main saffron production region of the world) in Iran. After digging up in early July (the true dormant period), healthy, wound-free and equal-size saffron corms were sorted (10–12 g). They were subsequently soaked into an acaricide (0.1% Propargite) and fungicide (0.15% Thiophanate-methyl) solution and dried for 1–2 h. The corms were incubated at 25 °C under dark conditions at a relative humidity of 85 ± 2% for 3 months, since previous studies showed that flower formation starts in this condition [47,48].

In early October, corms were stacked on metal mesh shelves and separately primed by GA$_3$ (Merck KGaA Co., Darmstadt, Germany) and GABA (Sigma-Aldrich Co., Darmstadt, Germany) standard solutions with 1 mM and 25 µM concentration, respectively, since previous literature showed that the highest growth was observed in plants receiving these concentrations [49,50]. Tween-20 (0.1%, *w/v*) was added as a surfactant to ensure the adhesion of PGRs to the treated corms. Spraying PGRs was repeated after 3 days. Then, corms were kept in a controlled phytotron (85% relative humidity, day/night temperature of 20 °C/15 °C with a photoperiod of 8 h day/16 h night, according to Fallahi and collaborators [51]). The irradiance was provided by LED modules with customized built spectra (12W, Parto Roshde Novin, Iran Grow Light, Tehran, Iran) at a photosynthetic photon flux density (PPFD) of 80 ± 10 µmol m$^{-2}$ s$^{-1}$. Four light qualities including white (wavelength 400–700 nm), monochromatic blue (peak wavelength at 465 nm), monochromatic red (peak wavelength at 660 nm), and a combination of blue and red lights (50% blue: 50% red) were set as light treatments (Figure 1). The light regimes were segregated using panels made of aluminum foil to avoid light transmission, and they were applied until the end of the flowering period. Light intensity and spectrum were checked by PAR-FluorPen FP 100-MAX (Photon Systems Instruments, Drasov, Czech Republic) and Sekonic C-7000 spectrometer (Sekonic Corp, Tokyo, Japan), respectively. For each treatment, 100 corms were used.

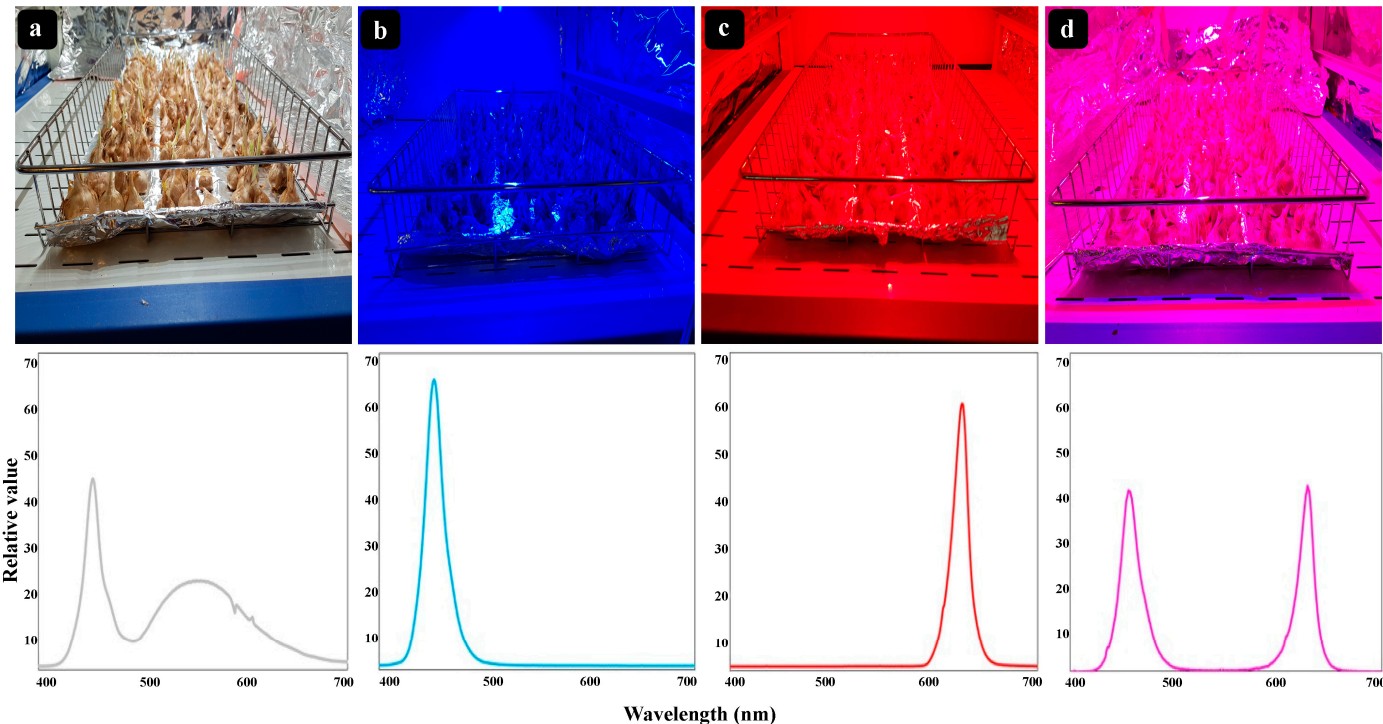

**Figure 1.** The lighting conditions together with the relative spectral distribution of each lighting treatment. Four lighting treatments are illustrated: White (**a**), monochromatic blue (**b**), monochromatic red (**c**), and a combination of blue and red lights (**d**). The light treatments were separated with a panel wall to segregate light regimes.

After 25 days from light treatments, corms were cold-shocked at 13 °C to induce flowering, something that is difficult to achieve in outdoor conditions where the temperature cannot be controlled.

### 2.2. Determining the Yield of Saffron Flower

Flowering was initiated 32–38 days after light treatments. During the flowering period, three plants were sampled for each replication. The flowers were harvested by hand and their fresh weights were recorded instantly after harvesting. Afterward, stigmas were pulled out from the flowers and their fresh weight was recorded immediately. The flowers and stigmas were weighed again after oven-drying (Binder, Tuttlingen, Germany) at 80 °C for 72 h to determine their dry weights [11]. Fresh and dry weights of flower and stigma were measured using a digital scale with a precision of 0.001 g (Mettler Toledo, Greifensee, Switzerland).

### 2.3. Determination of Phytochemical Compounds

To assay qualitative compounds in stigma, secondary metabolites of saffron stigmas were measured regarding ISO 3632-2 [52]. Accordingly, the dried stigmas were ground into a fine powder using a pestle and mortar, 0.5 g of which was transferred into a 1 L volumetric flask, and then brought to 1 L volume with distilled water. The mixture was stirred in the dark for 20 min using a magnetic stirrer. After filtration through quantitative filter paper (No131; Advantec Toyo Kaisha Co., Tokyo, Japan), the aqueous saffron extract was analyzed for spectral features using an UV-Vis spectrophotometer (Lambda 365, PerkinElmer, Waltham, MA, USA). Consistent with ISO, the absorbance of each aqueous solution was measured at a wavelength of 257, 330, and 440 nm to determine active

substances of picrocrocin, safranal, and crocin, respectively. The percentage of three types of qualitative compounds was calculated by the following equation:

$$E_{1cm}^{1\%} = \frac{A}{M} \times 100,$$ (1)

where *E* is the amount of specific qualitative compound in %; *A* is the obtained absorbance number at the relevant wavelength, and *M* is the stigma dry weight in milligrams.

### 2.4. Statistical Analysis

To test the normality of data distribution, Anderson-Darling criterion was used at a significance level of 0.05 using the program MINITAB, version 14.1 (Minitab LLC, State College, PA, USA). The measured data were statistically assessed by analysis of variance (ANOVA) according to the 'factorial experiment in a randomized complete block design' method using two PGRs and four different light treatments with three replications. The data mean with standard errors were compared using Duncan's multiple range test at the $p \leq 0.05$ probability level using the SAS software, version 9.4 (SAS Institute, Cary, NC, USA). Principal component analysis (PCA) was carried out using the prcomp command of the R statistical software [53]. Since variables had different units of measurement, PCA was performed based on the correlation matrix [54].

### 3. Results

### 3.1. The Floral Yield of Saffron Plants

While light quality and PGRs are effective factors influencing plant's growth and development, studies on their interactions on the flower yield and stigma quality of saffron plants are scarce. In the present research, to evaluate the interactions of light quality and PGRs on the growth of saffron flowers, three months after incubation in the dark at 25 °C, corms were incubated at 13 °C under the PGRs and light treatments to induce flowering. Flowers emerged 32–38 days after light treatments. The flower number per corm, flower, and stigma fresh weight and dry weight were evaluated. The impacts of treatments on the number of flowers per corm, flower, and stigma fresh weight and dry weight were significantly different (Table 1). Not only were the levels of both factors significantly different, but their interaction was also significant in all of the investigated traits, which implies the different effects of treatments on flower characteristics of saffron. The results showed that the applications of PGRs and different light qualities were effective treatments for enhancing the floral yield of saffron plants. The maximum number of flowers per corm, flower, and stigma fresh and dry weights were obtained in the saffron plants grown under GABA+BR treatment (Figure 2).

**Table 1.** Analysis of variance for the effect of treatments on flower characteristics of saffron.

| S.O.V. | df | MS | | | | |
|---|---|---|---|---|---|---|
| | | Number of Flowers per Corm | Flower Fresh Weight (mg per Corm) | Flower Dry Weight (mg per Corm) | Stigma Fresh Weight (mg per Corm) | Stigma Dry Weight (mg per Corm) |
| R | 2 | 0.054 ns | 225,864.111 ** | 7255.083 ** | 727.083 ** | 9.631 * |
| PGR | 2 | 2.629 ** | 212,712.694 ** | 6869.083 ** | 1787.25 ** | 134.156 ** |
| Light quality | 3 | 0.733 ** | 54,509.435 ** | 1801.657 ** | 586.25 ** | 75.117 ** |
| PGR×light quality | 6 | 0.175 * | 5998.102 ** | 201.38 ** | 119.25 * | 17.839 ** |
| E | 22 | 0.059 | 711.626 | 23.508 | 45.265 | 1.762 |
| CV (%) | – | 21 | 6.7 | 6.7 | 19.1 | 17.6 |

ns not significant. * and ** Significant at $p \leq 0.01$ and $p \leq 0.05$, respectively. S.O.V. = Source of Variation; df = Degrees of Freedom; MS = Mean Squares; R = Replication; PGR = Plant Growth Regulator; E = Error; CV = Coefficient of Variation.

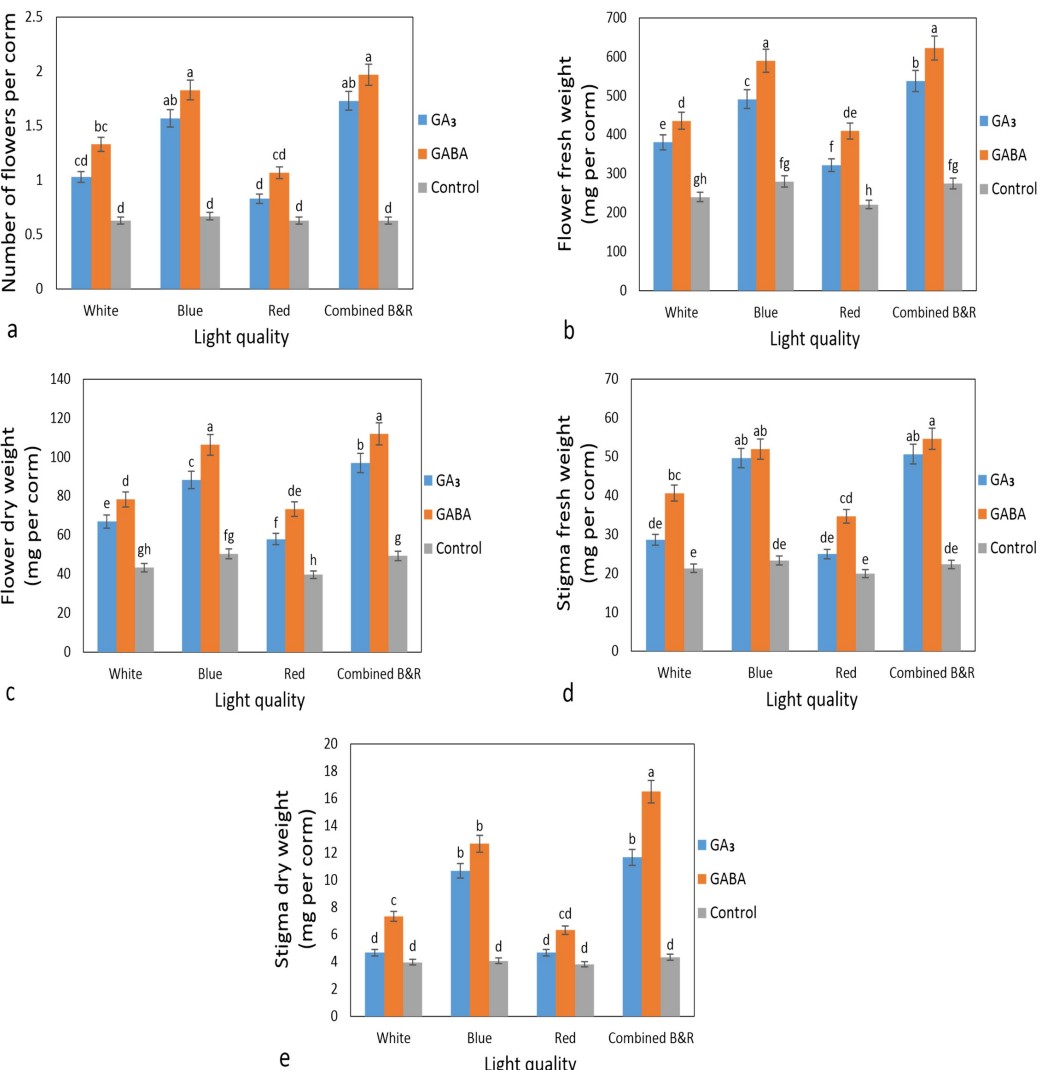

**Figure 2.** The number of flowers (**a**), flower fresh weight (**b**), flower dry weight (**c**), stigma fresh weight (**d**), and stigma dry weight (**e**) of saffron plants grown under two PGRs including GA$_3$ (gibberellic acid) and GABA (γ-aminobutyric acid), and four light qualities including white (W), blue (B), red (R) and combined B&R (50% blue: 50% red) (BR). Different letters indicate that values are significantly different at $p \leq 0.05$ according to Duncan's multiple range test. Error bars represent the mean value of three replications ± standard deviation.

The number of flowers in GABA+BR plants was almost three-fold higher than their numbers in plants grown under control treatments (i.e., control+W, control+B, control+R, and control+BR) (Figure 2a). No significant difference was noted regarding flower fresh and dry weights between GABA+BR and GABA+B. An intermediate flower fresh weight and dry weight was also detected under GABA+W and GABA+R. Fresh and dry weights of stigmas followed the same trend as of flowers. Stigma dry weight in GABA+BR saffron plants was significantly higher than the others. In this way, it was more than four times higher than its value in all control treatments regardless of the light spectrum.

### 3.2. The Phytochemicals Content

Phytochemical concentration in the flowers of saffron plants was analyzed to evaluate the influence of light quality and PGRs on its accumulation. According to the results, all characteristics were significantly affected by the different treatments (Table 2). The application of PGRs was the most effective treatment for inducing the production of secondary metabolites in the stigmas of saffron. The maximum amount of safranal (perfume) and

picrocrocin (flavor) was observed in GABA+BR treatment, with values of 5.03% and 15.8%, respectively (Figure 3a,b). The highest enhancement in crocin content (color) was recorded in stigmas of corms treated with GA₃+BR (Figure 3c). The concentration of safranal in flowers of GABA+BR was 2.1 times higher than its concentration in flowers of control+R. The results demonstrated that there is a positive interaction between PGRs and combined blue and red lights.

**Table 2.** Analysis of variance for the effect of treatments on phytochemical traits of saffron.

| S.O.V. | df | MS | | |
|---|---|---|---|---|
| | | Crocin (% DW) | Picrocrocin (% DW) | Safranal (% DW) |
| R | 2 | 84.011 ** | 13.792 ** | 39.531 ** |
| PGR | 2 | 128.661 ** | 10.262 ** | 3.491 ** |
| Light quality | 3 | 64.67 ** | 5.599 ** | 3.489 ** |
| PGR×light quality | 6 | 6.197 ** | 1.86 ** | 0.445 ** |
| E | 22 | 0.216 | 0.02 | 0.052 |
| CV (%) | − | 2.62 | 1.1 | 6.91 |

** Significant at $p \leq 0.05$. S.O.V. = Source of Variation; df = Degrees of Freedom; MS = Mean Squares; R = Replication; PGR = Plant Growth Regulator; E = Error; CV = Coefficient of Variation.

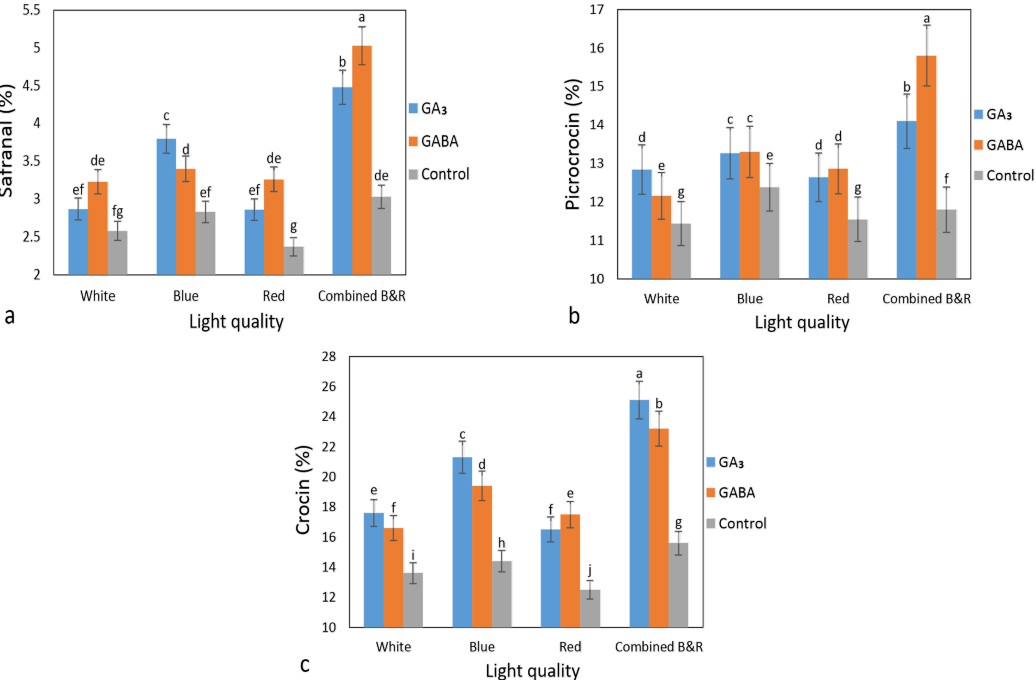

**Figure 3.** Safranal (**a**), picrocrocin (**b**), and crocin (**c**) accumulation of saffron flowers grown under two PGRs including GA₃ (gibberellic acid) and GABA (γ-aminobutyric acid), and four light quality including white, blue, red and combined B&R (50% blue: 50% red). Different letters indicate that values are significantly different at $p \leq 0.05$ according to Duncan's multiple range test. Error bars represent the mean value of three replications ± standard deviation.

In saffron, the concentration of the three major secondary metabolites is of great importance since it determines its quality [55]. Accordingly, PGRs+BR treatments had the highest efficiency among the treatments by improving the concentration of the main chemical ingredients.

### 3.3. Principal Component Analysis (PCA) of Variables

For an overall interpretation of the results obtained, PCA was used. The matrix of the analysis consisted of 12 cases corresponding to the treatments and eight variables

(measured characteristics). The first two PCs account for 92.5% and 4.3%, respectively, of the total variation in the dataset. Because of using the correlation matrix, two PCs are desirable, according to Jolliffe [54]. The results were presented as a two-dimensional scatter-plot of the data (Figure 4), which is, by definition, the best variance-preserving a two-dimensional plot of the data, representing over 96% of the total variation. The first PC separated the treatments control and red light from the others. The second PC separated the BR treatment from the other treatments.

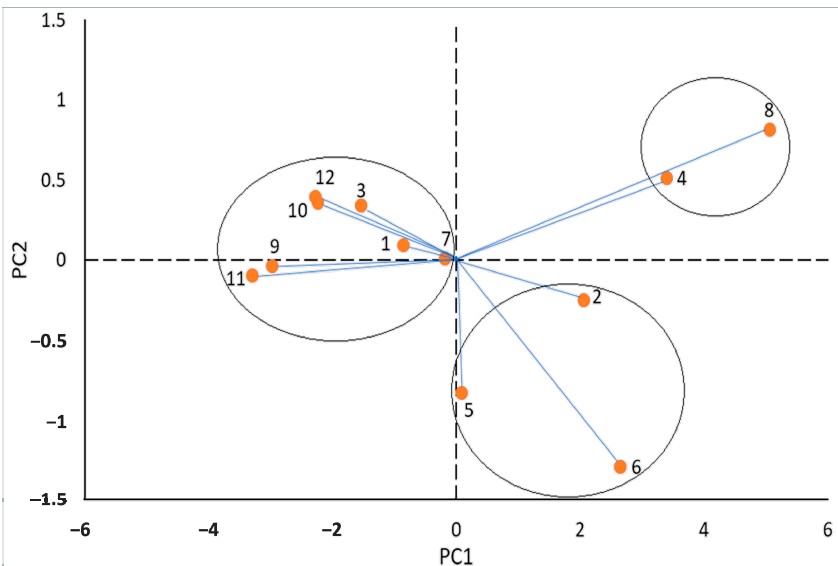

**Figure 4.** PCA biplot graphics of measured variables in saffron flowers grown under different conditions: 1 = GA$_3$+white light, 2 = GA$_3$+blue light, 3 = GA$_3$+red light, 4 = GA$_3$+combined B&R light, 5 = GABA+white light, 6 = GABA+blue light, 7 = GABA+red light, 8 = GABA+combined B&R light, 9 = control+white light, 10 = control+blue light, 11 = control+red light, and 12 = control+combined B&R light.

PCA biplot divided the treatments into three separate groups. The GA$_3$+BR treatment was associated with the GABA+BR treatment. These two treatments have the most positive effect on the yield and quality of saffron flowers, as previously shown (Figures 2 and 3), and the PC analysis could put them into one group. Finally, it can be largely claimed that the biplot graphic with the first two PCs was able to group the treatments well.

## 4. Discussion

The importance of light for plant growth and development is already well-known. Light quality is one of the important features of the lighting environment detected by very sophisticated sensory networks. Photomorphogenesis is a light-mediated development, where plant growth patterns respond to the light spectrum, direction and quantity [24,56,57]. Phytochromes, cryptochromes, and phototropins are photochromic sensory receptors that restrict the photomorphogenic effect of light to the UV-A, UV-B, blue, and red portions of the electromagnetic spectrum [58]. Light quality can also affect the synthesis of bioactive compounds and the secondary metabolism of plants [59]. Therefore, regulating light quality can be considered as a strategic tool to improve the yield of secondary metabolites [60].

PGRs are organic compounds that affect the physiological processes of plants. More specific responses include alteration of C partitioning, greater root-to-shoot ratios, enhanced photosynthesis, altered nutrient uptake, improved water status, and altered crop canopy [61]. PGRs play a significant role in the growth, development, and distribution of essential nutrients in plant systems [62]. PGRs can potentially play a fundamental role in regulating plant responses to various abiotic stresses and hence, contribute to plant adaptation under adverse environments [63].

In the present study, the floral yield and phytochemicals content of saffron plants were influenced by light quality and PGR application. PGRs are positive regulators of secondary metabolic pathways, and the highest improvement in phytochemicals content was observed in corms treated with PGRs. The results showed that the combination of 50% blue: 50% red light caused the generation of more flowers with the highest content of phytochemicals.

Certain quantitative and qualitative responses are expected when plants are grown under different light conditions. Literature data shows that a combination of blue and red light is of great importance for the normal growth and physiological performance of plants. For instance, the maximum photosynthetic capacity is reported in the leaves of cucumber plants grown under 50% red: 50% blue lights [64]. Klein and collaborators, and Naznin and collaborators found that mixed red and blue light led to higher Chlorophyll a, b and total Chlorophyll levels, increased electron transport rate (ETR) and an early onset of non-photochemical quenching (NPQ), all of which led to improvements in photosynthetic efficiency [65,66]. Recent experimental evidence indicates that the highest flower number and stigma fresh and dry weight of saffron plants were detected under monochromatic blue light, whereas no significant change was noted among monochromatic blue and 50% red: 50% blue lights [11]. Lettuce, spinach, kale, basil, and pepper plants grown under 100% red light showed lower fresh and dry mass compared to those cultured under a combination of red and blue lights [66]. The previous report has shown that mixed blue and red lights alter plant photomorphogenesis and photosynthesis, mainly through their effects on leaf anatomy, photosynthetic electron transport and the expression and activities of Calvin cycle-related enzymes, including Rubisco, FBPase and GAPDH [67]. Similar to our findings, Moradi and collaborators observed the lowest number of flowers as well as fresh and dry weight of flowers and stigmas in the saffron plants grown under monochromatic red light [11].

It is well known that light is a physical factor that can affect the composition of secondary metabolites. Kajikawa and collaborators found that a light environment with a low red to far-red (R/FR) photon flux density ratio increased the crocin concentration in the stigmas of hydroponically-cultivated saffron corms [68]. They assumed that the decrease in the R/FR ratio from 15.8 to 1.8 enhanced the accumulation of photosynthetic products that are sources of crocin biosynthesis. Sng and collaborators found that a combination of red and blue light induced the expression of genes involved in secondary metabolite biosynthesis [69].

GABA is a very important candidate for controlling a wide variety of physio-biochemical processes in plants. Regulations in cytosolic pH, anti-oxidative enzymatic systems, buffering agent in C and N metabolism, osmoregulation, armoring against oxidative stress, and signal transduction are the key roles of GABA that might improve overall plant performance [70]. Due to the different roles of GABA in a diverse range of processes, research on GABA can provide fundamental insight into various aspects of a plant's developmental system [31]. Fait and collaborators suggested that GABA is one of the main components providing C resources for the TCA (Tricarboxylic acid) cycle in tobacco plants [71]. In melon seedlings, a GABA-induced increase in nitrate absorption was associated with enhancing nitrate reductase activity [72]. There is evidence that $N_2$ fixation and nodule formation in legumes are regulated by GABA [73]. The indirect effect of GABA on the photochemical efficiency of lettuce plants by regulating the C:N ratio has been argued by Kalhor and collaborators [49]. Based on the results obtained by Vijayakumari and Puthur in black pepper plants, photosynthetic performance was improved by GABA exposure [74]. A growing body of evidence demonstrated the role of GABA as an endogenous plant signaling molecule and metabolite. According to the results of Hijaz and collaborators in *Citrus sinensis*, levels of plant hormones such as salicylic acid, benzoic acid, jasmonic acid, abscisic acid, and indole-3-acetic acid are significantly enhanced by GABA exposure [75]. In another study, the exogenous application of GABA led to the improvement in seedling growth and net photosynthesis of maize [70]. Being an essential intermediate of nitrogen

metabolism and amino acid biosynthesis, GABA plays a key role in primary and secondary metabolite synthesis [76]. The overall study by Kaur and Zhawar indicated GABA as an important regulator of secondary and carbohydrate metabolisms [77].

It has been shown that fluctuations in gibberellin concentration influence light-regulated seed germination, photomorphogenesis during de-etiolation, and photoperiod regulation of stem elongation and flowering [78]. It was reported that saffron corms treated with GA$_3$ have a stimulated decomposition of starch and an increased accumulation of soluble sugars, particularly sucrose [39]. They found that the accumulation of total pentoses and total ketoses was increased by GA$_3$. As starch breakdown is influenced by the sink demand, sucrose has been suggested to play a signal role in the systemic control of starch synthesis and breakdown [79]. GA$_3$ has been reported to increase the availability of carbohydrates in flowers, and since carbohydrates are the main source of nutrition for flowers [80], hormonal priming with GA$_3$ improved the yield of flowers and the quality of saffron stigma [81]. GA$_3$ affects the gene expression of invertase, and invertase activity in target tissues causes the production of hexoses (fructose and glucose), which are sources of carbon and energy for plant growth [82].

Mollafilabi reported that saffron produced under controlled conditions was graded as excellent according to Iranian national standards for stigma quality [83]. The author also stated that 1.38 flowers per corm were produced when saffron corms were kept under a controlled environment for 90 days during the flower initiation stage and then transferred to the flowering room during the flower emergence stage.

## 5. Conclusions

As the main chemical compounds, secondary metabolites play an important role in the saffron quality. The percentage of secondary metabolites determines the quality of saffron flowers. This study, for the first time, evaluated the interaction of PGRs and different light spectra on the flowering yield and phytochemical performance of saffron plants. Improvement of yield and quality of saffron in indoor farming was targeted in this study. According to the findings of the present study, the application of GABA with a combination of red and blue lights led to better flower production with a higher stigma yield. Additionally, this treatment also positively affected the contents of safranal and picrocrocin. Meanwhile, the application of GA$_3$ with a combination of red and blue lights led to the highest content of crocin. Ultimately, the utilization of PGRs and artificial light plays a crucial role in the enhancement of secondary metabolites and the improvement of physiological attributes of saffron. PCA was used for interpretation of the results and it confirmed our findings. The positive effect of PGRs on flowers number per corm, flower and stigma fresh weight and dry weight was determined for saffron plants. This experiment revealed that quality of stigma is influenced by the exogenous application of PGRs, especially GABA. Red and blue light spectra are necessary for plant photomorphogenesis, and their combination led to better yield and quality of saffron flowers.

Overall, the results of this research set up the best production protocol for saffron, and developing this protocol guarantees optimized conditions to produce top-quality saffron indoors.

**Supplementary Materials:** The following are available online at https://www.mdpi.com/article/10.3390/horticulturae9020169/s1, Table S1. Experimental design for arranging the treatments in saffron.

**Author Contributions:** Conceptualization, M.E. and M.G.J.; methodology, M.G.J. and S.A.; formal analysis, M.E.; investigation, M.E.; data curation, M.E.; writing—original draft preparation, M.E.; writing—review and editing, M.G.J., S.A. and S.N.; supervision, M.G.J. and S.A.; project administration, M.G.J.; funding acquisition, S.N. All authors have read and agreed to the published version of the manuscript.

**Funding:** This research was funded by the University of Tehran and partly by the Italian Ministry of Education and Research, Project MUR - PRIN 2020, Ref. 2020ELWM82.

**Institutional Review Board Statement:** Not applicable.

**Informed Consent Statement:** Not applicable.

**Data Availability Statement:** The data presented in this study are available on request from the corresponding authors. The data are not public.

**Acknowledgments:** The authors would like to thank Hamid Ahadi for his technical support in the phytochemical analyses.

**Conflicts of Interest:** The authors declare no conflict of interest.

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
