# Peer review of "Alteration of Flower Yield and Phytochemical Compounds of Saffron (Crocus sativus L.) by Application of Different Light Qualities and Growth Regulators"

_horticulturae, doi:10.3390/horticulturae9020169_

Round 1

Reviewer 1 Report

Line 24: What kind of quality? commercial, nutritional, therapeutic? Please, specify

Line 24: What chemical ingredients are you referring to?

Lines 39-47: Please add references

Line 76: Red and blue spectra has previously been studied for saffron production? What is the difference between those studies and your proposal?

Line 97-105: I suggest you not go back to information from paragraph 3 in the last paragraph, but rather highlight your objective.

Line 117: Was the control just water? Please, specify

Line 121: Based on what characteristics did you select the saffron corms? Please, specify

Line 127: Was the added volume of PGRs standardized? Please, specify

Line 128: How did you select the concentrations of the PGRs? Is there previous literature? or a previous screening analysis?

Line 150: Please, remove –

Author Response

Dear Reviewer

Thank you for the well recommendations and points to improve the quality of our manuscript.

According to the reviewer’s suggestion, we revised some words and terms in the text.

Answers to questions:

Line 24: What kind of quality? commercial, nutritional, therapeutic? Please, specify

Response: Commercial and pharmaceutical quality.

Now specified in line 24.

Line 24: What chemical ingredients are you referring to?

Response: Saffron's chemical ingredients (compounds).

Now specified in line 24.

Lines 39-47: Please add references

Response: Two references were added in lines 41 and 45.

Line 76: Red and blue spectra has previously been studied for saffron production? What is the difference between those studies and your proposal?

Response: The three biggest differences are as follows:

First, corms, in previous studies, have been planted in pots, while our experiment was accomplished fogponically;

Second, our experiment, for the first time, determined the effect of GABA;

and third, the interaction of PGRs and different light qualities were evaluated as well.

This information has been provided in lines 16-20 and 119-126.

Line 97-105: I suggest you not go back to information from paragraph 3 in the last paragraph, but rather highlight your objective.

Response: We modified according to your request, although we thought to keep some information in the last paragraph for integrating the preceding information so that the importance of the present study be clarified. The modifications are visible now in lines 75-78 and 107-109.

Line 117: Was the control just water? Please, specify

Response: Control was referring to untreated corms and we treated with tap water.

We added now in line 132.

Line 121: Based on what characteristics did you select the saffron corms? Please, specify

Response: Healthy, wound-free and equal-sized corms weighing between 10-12 g were selected.

We added the information now in line 136.

Line 127: Was the added volume of PGRs standardized? Please, specify

Response: Standard solutions were used.

Now specified in lines 142,143.

Line 128: How did you select the concentrations of the PGRs? Is there previous literature? or a previous screening analysis?

Response: The concentrations were selected based on the results of previous studies:

  1. Kalhor, M.S.; Aliniaeifrad, S.; Seif, M.; Asayesh, E.J.; Bernard, F.; Hassani, B.; Li, T. Enhanced salt tolerance and photosynthetic performance: implication of γ-amino butyric acid application in salt-exposed lettuce (Lactuca sativa L.) plants. Plant Physiol. Biochem. 2018, 130, 157–172. [https://doi.org/10.1016/j.plaphy.2018.07.003]
  2. Mzabri, I.; Rimani, M.; Kouddane, N.; Berrichi, A. Study of the effect of pretreatment of corms by different concentrations of gibberellic acid and at different periods on the growth, flowering, and quality of saffron in eastern Morocco. Adv. Agric. 2021, 2021, 9979827. [https://doi.org/10.1155/2021/9979827]

We added the information now in lines 143-145.

Line 150: Please, remove –

Response: We removed – now in line 168.

NB: References have been arranged throughout the manuscript according to the new numbering.

Reviewer 2 Report

Abstract

The specific data in this research should be stated in this section.

Introduciton

1、 Line 37-38, please give the specific wordwide yield of Saffron.

2、 Line 63. They reported that saffron production was more favorable under the soilless production system than the traditional production in soil. Please specify how favorable?

3、 Line 91. Since GA3 regulates growth, applications of very low concentrations can have a profound effect…. What is the range of low concentrations?

4、 Line 101. The lighting spectra has been shown to affect the flower production of saffron…please specify how?

5、 Line 108-111. Please specify the content of this research.

Materials and Methods

1、 The design of this experiment should be given, you can add a table to show it.

2、 The number of coordinates in Figure 1 is not clear, please improve the quality of this figure.

3、 Line 128. Tween-20 (0.1%, W/V)

4、 Line 158, 50 °C for 72 h? Under vacuum or under atmospheric pressure? I don’t think the material can be dried to a constant weight, please give the reference.

Results

1、 The units should be given in Table 1 and Table 2.

2、 Table 1 is not discussed enough

3、 Figure 2 is not clear

Author Response

Dear Reviewer

Thank you for the well recommendation and the points to improve the quality of the manuscript.

According to the reviewer’s suggestion, we revised some words and terms in the text.

Answer to questions:

Abstract

The specific data in this research should be stated in this section.

Response: Thank you for pointing out missing data. We added specific data in the abstract, now in lines 22,25,26.

Introduction

1、 Line 37-38, please give the specific worldwide yield of Saffron.

Response: The global yield of saffron under open field conditions is estimated at 3.4 kg ha-1, according to Cardone et al. (2020).

[https://doi.org/10.1016/j.scienta.2020.109560]

We added the specific information now in lines 38,39.

2、 Line 63. They reported that saffron production was more favorable under the soilless production system than the traditional production in soil. Please specify how favorable?

Response: In the first and the second flowering seasons, the percentage of flowering corms was higher under the soilless production system (39 and 65%, respectively) than the traditional production in soil (6 and 56%, respectively).

We added the information now in lines 67-70.

3、 Line 91. Since GA3 regulates growth, applications of very low concentrations can have a profound effect…. What is the range of low concentrations?

Response: Less than 250 ppm, according to Keshtkar, H.R. et al. (2008). Seed dormancy-breaking and germination requirements of Ferula ovina and Ferula gummosa. Desert, 13: 45-51.

We added this information now in line 101.

4、 Line 101. The lighting spectra has been shown to affect the flower production of saffron…please specify how?

Response: Through altering biomass partitioning toward flowers.

We added this comment now in line 112.

5、 Line 108-111. Please specify the content of this research.

Response: We added information now in lines 122-126.

Materials and Methods

1、 The design of this experiment should be given, you can add a table to show it.

Response: Thank you for your request. We added the table of the experimental design, although we thought that the design of an experiment can be given as a supplementary material. Thus we added in the supplementary material as Table S1.

2、 The number of coordinates in Figure 1 is not clear, please improve the quality of this figure.

Response: A high-resolution picture was replaced. We add the original pictures as separated files as well.

3、 Line 128. Tween-20 (0.1%, W/V)

Response: We added ", W/V" now in line 147.

4、 Line 158, 50 °C for 72 h? Under vacuum or under atmospheric pressure? I don’t think the material can be dried to a constant weight, please give the reference.

Response: That was a misspelling; 80 °C for 72 h is correct, according to Moradi et al. (2021).

[https://doi.org/10.3390/cells10081994]

We corrected and added the reference now in line 178.

Results

1、 The units should be given in Table 1 and Table 2.

Response: We added the units in the tables.

2、 Table 1 is not discussed enough

Response: We added further discussion. Now in lines 217-221.

3、 Figure 2 is not clear

Response: We changed with the highest resolution graphs. We add the original figures as separate files as well.

NB: References have been arranged throughout the manuscript according to the new numbering.

Reviewer 3 Report

Dear authors, the suggestions attached are intended to highlight more the quality and the importance of your results.

Author Response

Dear Reviewer

Thank you for the fine recommendations and points to improve the quality of our manuscript.

According to the reviewer’s suggestion, we revised some words and terms in the text.

Answer to questions:

Dear authors, the suggestions attached are intended to highlight more the quality and the importance of your results.

Suggestions for the authors:

Line 52 – Please find a reference related to LED influence on saffron cultivation, if not available better skip it, creates confusions talking about watermelon grafting. Another option can be to move massive info in the Introduction section info as suggested below at Discussions.

Response: Thank you very much for your suggestions. We removed the reference on watermelon and used instead one on saffron cultivation [https://doi.org/10.3390/cells10081994]

The changes are now in lines 50-56.

Lines 62, 132, 298, 299, 311, 315, 319, 328, 338 – Better replace colleagues with collaborators

Response: 11 replacements done throughout the manuscript.

Table 1 – Please be so kind and clarify if Light quality df 3, (3 also in the title) according to the lines 134 - 137 Four light qualities including white (wavelength 400-700 nm), monochromatic blue (peak wavelength at 465 nm), monochromatic red (peak wavelength at 660 nm), and a combination of blue and red lights (50% blue: 50% red) were set as light treatments

Response: Factor “light” had four levels (spectra): 1) white, 2) blue, 3) red, 4) blue+red. Thus, in the anova we had 4 n and 3 df for the light factor. 

Discussions – There are provided a lot of references related to the different species responses to the different light conditions and GABA applications. I personally consider this information it fits well in INTRODUCTION section and here must be kept only references related to saffron. In this manner the output of the current study is better understanded and easy to be followed.

Response: Thank you very much for your comment. Despite the following publication (https://doi.org/10.3390/cells10081994) on saffron, there is little information related to the application of light qualities and GABA on the species, thus we thought to provide a comprehensive discussion comparing the effects of the treatments on other species. When there is not enough information regarding the effects of experimental treatments on the studied species we need to rely on similar approaches. We trust that research on saffron will continue around the world so we will have more specific information, but at the moment we think it is necessary to discuss the novelty on saffron compared to the state of the art.

We hope the review is satisfied for the explanation, otherwise we might reconsider if you feel strong about this aspect.

Conclusions – Lines 369 to 374 the info reflects the feature of the control cultivation system in general and presents talks about disorders and diseases which are not subject to the current work; also the reference to the interest of restaurants and companies can be considered a suppositions, not being strictly related to the output of the current research, better remove the first paragraph.

Response: Thank you very much for your considerations toward which we do agree; we have removed those parts (now lines 398-404).

NB: References have been arranged throughout the manuscript according to the new numbering.

Reviewer 4 Report

Dear authors,

The publication is interesting work. The conclusion must be improved, adding the numbers of results. 

Best regards,

Author Response

Dear Reviewer

Thank you for the recommendations and points to improve our manuscript.

According to the reviewer’s suggestion, we revised some words and terms in the text.

Comments and Suggestions for Authors:

Dear authors,

The publication is interesting work. The conclusion must be improved, adding the numbers of results. 

Best regards,

Response: Further results have been provided in the "conclusions" section. Now in lines 403-420.

NB: References have been arranged throughout the manuscript according to the new numbering.

Round 2

Reviewer 2 Report

This papercould be accepted in present form。